# Nonlinear Control System Design of an Underactuated Robot Based on Operator Theory and Isomorphism Scheme

**Mingcong Deng**  **and Shotaro Kubota**

The Graduate School of Engineering, Tokyo University of Agriculture and Technology, 2-24-16 Nakacho, Koganei, Tokyo 184-8588, Japan; 1212kubota@gmail.com

\* Correspondence: deng@cc.tuat.ac.jp

**Abstract:** The number of actuators of an underactuated robot is less than its degree of freedom. In other words, underactuated robots can be designed with fewer actuators than fully actuated ones. Although an underactuated robot is more complex than a fully actuated robot, it has many advantages, such as energy, material, and space saving. Therefore, it has high research value in both control theory and practical applications. Swing-up is a mechanism with two links, which mimics a gymnast performing a horizontal bar movement. Over the past few decades, many sufficiently robust control techniques have been developed for a fully actuated robot but almost none of them can be directly applicable to an underactuated robot system. The reason is that such control techniques require certain assumptions that are valid only for fully actuated robot systems but not for underactuated ones. In this paper, a control system design method for underactuated robots based on operator theory and an isomorphism scheme is first proposed. Bezout identity is designed using isomorphism. The effectiveness of the design method is confirmed by simulation. The simulation results show that the performances, such as robust stability and response time, of an underactuated robot control system are improved.

**Keywords:** nonlinear control system; operator theory; right coprime factorization; underactuated robot; swing-up

**MSC:** 93C10

## 1. Introduction

According to the relationship between degree of freedom (DOF) of a robot and the number of independent control inputs, robots can be devided into three types: fully actuated, redundantly actuated, and underactuated ones [1]. The number of actuators of an underactuated robot is less than its DOF. Despite the higher complicity of an underactuated robot than that of a fully actuated one , it has the advantages of energy, material, and space saving, etc. In some specific cases, if there is precise drive control, it can achieve higher efficiency and better flexibility despite its high DOF [2]. Therefore, it deserves more investigations in theory and practice.

Its typical example is acrobot. An acrobotic robot is a two-link mechanism that mimics a gymnast performing a horizontal bar movement [3]. In recent decades, many sufficiently robust control techniques have been developed for a fully actuated robot but they are not directly applicable to an underactuated robot system. [4] concerns the energy-based swing-up control for a remotely driven acrobot (RDA), which is a 2-link planar robot with the first link being underactuated and the second link being remotely driven by an actuator mounted at a fixed base through a belt. An energy-based control law for swinging up the acrobot is proposed in [5]. The control law is designed and the convergence analysis is carried out based on Lyapunov stability theory. The paper [6] provides a complete analysis of the convergence of the energy and the motion of the acrobot and clearly illustrates

several unique characteristics of the closed-loop system of the acrobot under the energy-based control. The combination of the partial linearization control for the swing-up phase proposed by [7] and the robust control for the capture and balance phase is utilized in [8]. The reason is that their deployment requires some assumptions that are true for fully actuated robot systems but not for an underactuated one [3,9]. A major problem is that the stability of the control system cannot be guaranteed when they are applied to the latter. The stabilization analysis in [10] is based on the attractive ellipsoid method (AEM) for a class of uncertain nonlinear systems having "quasi-Lipschitz" nonlinearities. The paper [11] presents a development of adaptive state estimator and output controller based on Attractive Ellipsoid Method (AEM) for the stabilization of the Furuta's pendulum. The proposed method guarantees that the controlled system trajectories are stabilized within an ellipsoid of a "minimal size". The proposed method for swinging up and stabilizing underactuated two-link robots in [12] does not need to switch control laws when the system is near to the desired equilibrium point, and as the system approach to this equilibrium, the nonlinear control law becomes an LQR controller. Moreover, nonlinear dynamics is also a factor that needs to be considered. In [13], Zakai and Kushner-Stratonovich equations of the nonlinear filtering problem for a non-Gaussian signal-observation system are considered. Operator theory is a kind of nonlinear control theory which has the characteristics of nonlinear and uncertain unstable elements [14–20]. In addition, robust stability analysis can be performed only in a time domain without conversion to a frequency domain. Its advantage is that it can be done relatively easily, which indicates the effectiveness of operator theory. Further, an isomorphism is a map from one algebraic structure to another of the same type that preserves some relevant structures and properties uch as identity elements, inverse elements, and binary operations [21]. That is, the robust stability for nonlinear feedback control systems and the output tracking problems are studied. For the problem of factorizing an unstable plant for its nonlinear feedback control systems, its right factorization can be conducted by using an isomorphism approach [22].

In this paper, a control system design method based on operator theory and an isomorphism scheme is proposed to improve the performance of an underactuated robot with instability and uncertainties. That is, operator theory is employed for guaranteeing the robust stability, while an isomorphism scheme is used to avoid the existence of differential controller in operator-based control systems. The application of operator theory and the isomorphism scheme enable a shorter swing-up time. Extensive simulation is performed to validate the effectiveness of the proposed method. In summary, the highlights of our work include: (1) We use isomorphism to design feedback controller $Q^{-1}$ to stabilize the system for the first time. (2) We use the new stable term to design Bezout identity. (3) We have applied it in acrobot and obtained good results.

The remainder of this paper is organized as follows. In Section 3, modelling of swing-up and operator theory are introduced. In Section 4, a control system design method for an underactuated robot is presented. The simulations are given to illustrate the effectiveness of the proposed method in Section 5. Finally, conclusions are drawn in Section 6.

## 2. Notation

In this section, we will present some notations in this paper in Table 1.

**Table 1.** Notations.

| | |
|---|---|
| $\mathcal{S}(U,Y)$ | the set of stable operators from $U$ to $Y$ |
| $\mathcal{U}(U,Y)$ | the set of unimodular operators |
| $\|\cdot\|_{U_s}, \|\cdot\|_{Y_S}$ | norm |
| $\|A\|$ | Lipschitz semi-norm |
| $A : U_S \to Y_S$ | an operator mapping from $U_s$ to $Y_S$ |
| $\mathcal{D}(A)$ | the domain and range of $A$ |
| $\mathcal{R}(A)$ | the range of $A$ |
| $\mathcal{N}(U_s; Y_s)$ | the family of all nonlinear operators mapping from $\mathcal{D}(A) \subseteq U_s$ into $Y_s$ |
| $U_s, Y_s$ | normed linear space over the field of complex numbers endowed with norms $\|\cdot\|_{U_s}, \|\cdot\|_{Y_S}$ |
| $\mathrm{Lip}(D_S, Y_s)$ | a Lipschitz operator mapping from $D_s$ to $Y_s$ |
| $P$ | plant |
| $\Delta P$ | uncertainties |
| $\tilde{P}$ | the actual plant with uncertainties |
| $A, N, B, D, \tilde{A}, \tilde{N}, \tilde{B}, \widetilde{D}$ | operators of the system |
| $M, \tilde{M}$ | unimodular operators |

## 3. Preliminaries

### 3.1. Modeling of Swing-Up

As mentioned above, swing-up is a two-link device that mimics a gymnast performing a horizontal bar movement, as shown in Figure 1. Link1 models a gymnast's hand, and Link2 models a gymnast's waist [9]. Driving torque is applied only to the Link1, and the system can rotate freely. Swing-up has only one actuator to drive, but there are two links to be controlled. Thus, it is an underactuated robot system. Table 2 shows each parameter. Then, the equations of motion for each link are derived using the Lagrange equations of motion. Therefore, the motion equations of swing-up are as follows

$$d_{11}\ddot{q}_1 + d_{12}\ddot{q}_2 + h_1 + \phi_1 = 0 \tag{1}$$

$$d_{21}\ddot{q}_1 + d_{22}\ddot{q}_2 + h_2 + \phi_2 = \tau \tag{2}$$

where $d_{11}, d_{12}, d_{21}, d_{22}, h_1, h_2, \phi_1,$ and $\phi_2$ are shown in the following equations

$$d_{11} = m_1 l_{c1}^2 + m_2(l_1^2 + l_{c2}^2) + 2l_1 l_{c1} \cos q_2 + I_1 + I_2$$

$$d_{12} = d_{21} = m_2(l_{c2}^2 + l_1 l_{c2} \cos q_2) + I_2$$

$$d_{22} = m_2 l_{c2}^2 + I_2$$

$$h_1 = -m_2 l_1 l_{c2} \dot{q}_2^2 \sin q_2 - 2m_2 l_2 l_{c2} \dot{q}_1 \dot{q}_2 \sin q_2$$

$$h_2 = m_2 l_1 l_{c2} \dot{q}_1^2 \sin q_2$$

$$\phi_1 = (m_1 l_{c1} + m_2 l_1)g \cos q_1 + m_2 l_{c2} g \cos(q_1 + q_2)$$

$$\phi_2 = m_2 l_{c2} g \cos(q_1 + q_2)$$

**Table 2.** Parameter of swing-up.

| | | |
|---|---|---|
| $L$ | Lagrangian | [J] |
| $K$ | Kinetic energy of acrobot | [J] |
| $V$ | Potential energy of acrobot | [J] |
| $q_i$ | Target angle | [rad] |
| $\tau$ | Torque | [N·m] |
| $q_1$ | Angle of Link1 | [rad] |
| $q_2$ | Angle of Link2 | [rad] |
| $m_1$ | Mass of Link1 | 0.175 kg |
| $m_2$ | Mass of Link2 | 0.285 kg |
| $l_1$ | Length of Link1 | 0.3 m |
| $l_2$ | Length of Link2 | 0.5 m |
| $l_{c1}$ | Lengh from First joint to the center of gravity of Link1 | 0.177 m |
| $l_{c2}$ | Lengh from Second joint to the center of gravity of Link2 | 0.25 m |
| $I_1$ | Moment of inertia of Link1 | 0.0013 kg·m$^2$ |
| $I_2$ | Moment of inertia of Link2 | 0.0059 kg·m$^2$ |
| $g$ | Acceleration of gravity | [m/s$^2$] |

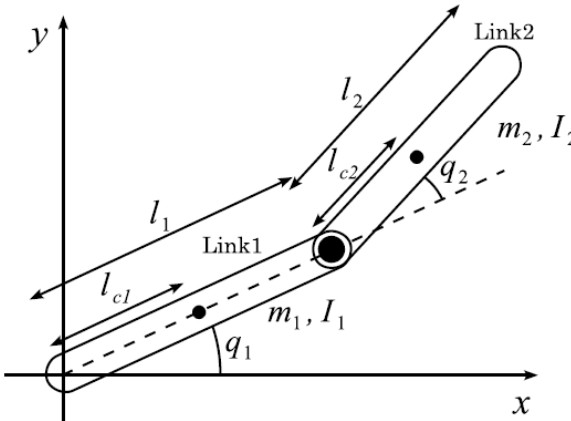

**Figure 1.** Model of swing-up.

### 3.2. Operator Theory

In this section, operator theory [14] is described. Using operator theory, a robust stability analysis can be performed even with uncertainties that are difficult to model in mathematics. Here, a nonlinear control system with uncertainties can be designed in the time domain by operator theory instead of being converted into the frequency domain by using a transfer function in a linear system. A nonlinear feedback system based on operator theory is shown in Figure 2.

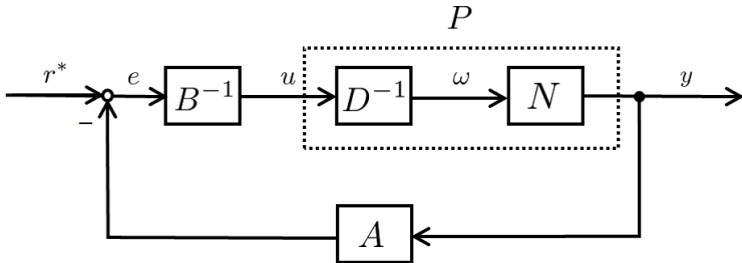

**Figure 2.** Nonlinear feedback system based on operator theory.

**Definition 1.** *Let $\mathcal{S}(U,Y)$ be the set of stable operators from $U$ to $Y$. Then, $\mathcal{S}(U,Y)$ contains a subset defined by*

$$\mathcal{U}(U,Y) = \left\{ M \in \mathcal{S}(U,Y); M \text{ is invertible with } M^{-1} \in S(Y,U) \right\}$$

*Elements of $\mathcal{U}(U,Y)$ are called unimodular operators.*

Let $U_s$ and $Y_s$ be two normed linear spaces over the field of of complex numbers, endowed, respectively, with norms $\|\cdot\|_{U_s}$ and $\|\cdot\|_{Y_s}$. Let $A : U_s \to Y_s$ be an operator mapping from $U_s$ to $Y_s$ and denoted by $D(A)$ and $R(A)$, respectively, the domain and range of $A$. Let $N(U_s; Y_s)$ be the family of all nonlinear operators mapping from $D(A) \subseteq U_s$ into $Y_s$. Let $D_s$ be a subset of $U_s$ and $F(D_s, Y_s)$ be the family of operators $A$ in $N(U_s, Y_s)$ with $D(A) = D_s$. A (semi)-norm on (a subset of) $F(D_s, Y_s)$ is denoted by

$$\|A\| := \sup_{\substack{x_1, x_2 \in D_S \\ x_1 \neq x_2}} \frac{\|A(x_1) - A(x_2)\|_{Y_s}}{\|x_1 - x_2\|_{U_S}}$$

if it is finite. In general, it is a semi-norm in the sense that $\|A\| = 0$ does not necessarily imply $A = 0$. In fact, it can be easily seen that $\|A\| = 0$ if and only if $A$ is a constant operator (need not be zero) that maps all elements from $D_s$ to the same element in $Y_s$.

**Definition 2.** *Let $\mathrm{Lip}(D_S, Y_S)$ be the subset of $\mathcal{F}(D_s, Y_s)$ with each element $A$ satisfying $\|A\| < \infty$. Each $A \in \mathrm{Lip}(D_S, Y_S)$ is called a Lipschitz operator mapping from $D_s$ to $Y_s$, and the number $\|A\|$ is called the Lipschitz seminorm of the operator $A$ on $D_s$.*

(i) **Right factorization:** Let the input space be denoted by $U$ and output space by $Y$. In general, these spaces are different extended linear spaces. Let the plant operator $P : U \to Y$ be such that $y(t) = P(u(t))$ where $u(t) \in U$ and $y(t) \in Y$. In addition, let $W$ be an auxiliary linear space and let the operator $N \to Y$ be stable such that $N(w(t)) = y(t)$, $w(t) \in W$, and let $D : W \to U$ be stable and invertible such that $D(w(t)) = u(t)$. It follows that the plant $P$ has a right factorization determined by $N$ and $D^{-1}$

$$P = ND^{-1} \tag{3}$$

(ii) **Right coprime factorization:** Suppose there is a right factorization operator $N, D$ in plant $P$. The Bezout equation is obtained as

$$AN + BD = M, \exists M \in \mathcal{U}(W, U) \tag{4}$$

If a stable operator $A$ and a stable and inversible operator $B$ satisfy the above Bezout Equation (4), then, $A$, $B$, $N$, and $D$ are the right coprime factorization of plant $P$ (as shown in Figure 2). At this time, the stability of the control system can be guaranteed.

(iii) **Robust right coprime factorization:** In general, there are uncertainties that are difficult to express in a mathematical model in an actual nonlinear control system. Thus, the nonlinear control system with uncertainties may be unstable. Using robust right coprime factorization to factorize a plant can guarantee the robust stability of a nonlinear feedback system with uncertainties. The nonlinear feedback system with

uncertainties is shown in Figure 3. Plant $P$ without uncertainties is the nominal plant, and the actual plant with uncertainties $\Delta P$ is $\tilde{P} = P + \Delta P$

$$\tilde{P} = P + \Delta P = (N + \Delta N)D^{-1} \tag{5}$$

$$A(N + \Delta N) + BD = \tilde{M} \tag{6}$$

$$A(N + \Delta N) + BD = AN + BD = M \tag{7}$$

$$\|(A(N + \Delta N) - AN)M^{-1}\|_{Lip} < 1 \tag{8}$$

where $\tilde{M}$ is unimodular. The nonlinear feedback system with uncertainties can be robust stable if (8) is satisfied.

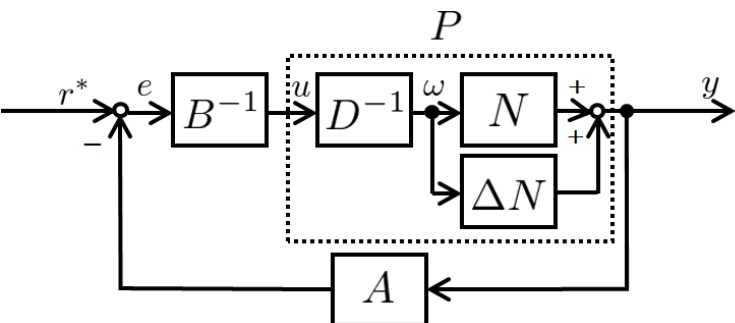

**Figure 3.** Nonlinear feedback control system with uncertainties.

## 4. Nonlinear Control System Design

*4.1. Tracking Controller Design of Swing-Up*

In this section, we explain the method for determining the target angle and the tracking controller design scheme in the swing-up control system.

### 4.1.1. Determining Target Angle

The swing-up control can make the second link track the target angle $q_2^d$ in the direction in which the energy of the first link is amplified. Therefore, the target angle $q_2^d$ is a time-varying value that changes depending on the state of each link, and it is necessary to change the target angle at any time. If the angle of the second link does not become 0 near the inverted point, it is difficult to stabilize the inverted state. $q_2^d$ is designed as

$$q_2^d = tan^{-1}((1 - \cos q_1)\dot{q}_1) \tag{9}$$

### 4.1.2. Tracking Controller Design

In this section, in order to make $q_2$ track $q_2^d$, we design a tracking controller $C$ as shown in Figure 4. The design method of the tracking controller is as follows.
First, from (1) and (2), the following equation can be obtained.

$$\bar{d}_2 \ddot{q}_2 + \bar{h}_2 + \bar{\phi}_2 = \tau \tag{10}$$

The definitions of $\bar{d}_2$, $\bar{h}_2$ and $\bar{\phi}_2$ are shown in Equations (11)–(13).

$$\bar{d}_2 = d_{22} - d_{21}d_{12}/d_{11} \tag{11}$$

$$\bar{h}_2 = d_{21}h_1/d_{11} \tag{12}$$

$$\bar{\phi}_2 = d_{21}\phi_1/d_{11} \tag{13}$$

$$\ddot{q}_2 = v_2 \tag{14}$$

$x_1 = q_2$ , $x_2 = \dot{q}_2$, so,

$$\begin{bmatrix} \dot{x}_1 \\ \dot{x}_2 \end{bmatrix} = \begin{bmatrix} 0 & 1 \\ 0 & 0 \end{bmatrix} \begin{bmatrix} x_1 \\ x_2 \end{bmatrix} + \begin{bmatrix} 0 \\ 1 \end{bmatrix} v_2$$

where $v_2$ can be expressed as

$$v_2 = -k_{\mathrm{p}}(q_2 - q_2^d) - k_{\mathrm{d}}\dot{q}_2 \tag{15}$$

As a result, the swing-up control is performed when an input $q_2 - q_2^d$ approaches 0. Here, $k_{\mathrm{p}}$ and $k_{\mathrm{d}}$ are design parameters.

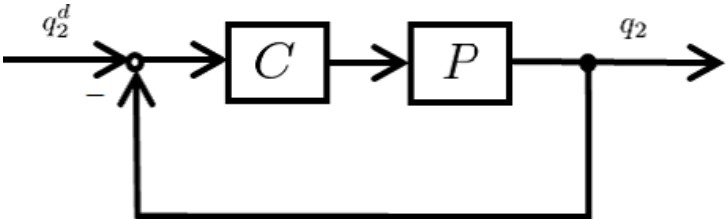

**Figure 4.** Feedback system using tracking controller.

### 4.2. Control System Design Based on Operator Theory and Isomorphism Scheme

Since there are unstable elements in the model equation of the swing-up, the stability of the control system cannot be guaranteed. In this section, the swing-up controller design method using operator theory and isomorphism scheme is proposed.

#### 4.2.1. Right Factorization of the Swing-Up

In actual nonlinear control system, it is common that there are uncertainties in swing-up, which uncertainties are difficult to to be modelled. The uncertainties can be seen as instability elements, which destabilize the system. Here, the instability elements are aggregated in $D^{-1}$. Right factorization of the swing-up is given by

$$D(\omega)(t) = \left( c_2 - \frac{(c_2 + c_3 \cos \omega(t))^2}{c_1 + c_2 + 2c_3 \cos \omega(t)} \right) \ddot{\omega}(t)$$
$$+ \frac{(c_2 + c_3 \cos \omega(t))(-c_3 \dot{\omega}^2(t) \sin \omega(t) - 2c_3 \dot{\omega}(t)\dot{q}_1 \sin \omega(t))}{c_1 + c_2 + 2c_3 \cos \omega(t)}$$
$$+ \frac{(c_2 + c_3 \cos \omega(t))(c_4 \sin q_1 + c_5 \sin(q_1 + \omega(t)))}{c_1 + c_2 + 2c_3 \cos \omega(t)} \tag{16}$$

and $\hspace{6cm}$ (17)

$$N(\omega)(t) = \omega(t)$$

where

$$c_1 = m_1 l_{c1}^2 + m_2 l_1^2 + I_1$$
$$c_2 = m_2 l_{c2}^2 + I_2$$
$$c_3 = m_2 l^2 l_{c2}$$
$$c_4 = (m_1 l_{c1} + m_2 l_1)g$$
$$c_5 = m_2 l_{c2}g$$

#### 4.2.2. Right Coprime Factorization of Underactuated Robot

When the operators $A$ and $B$ are designed so that $M$ is unimodular, and the Bezout equation $AN + BD = M$ is satisfied, then the operators $D^{-1}$ and $N$ form a right coprime factorization of the plant operator $P$, and the corresponding nonlinear feedback system is

shown in Figure 5. Bounded input and bounded output (BIBO) stability are guaranteed. Operators $A$ and $B$ are designed as follows using a design parameter $K$

$$A(y)(t) = (M - KD)(\omega)(t) \tag{18}$$

$$B(u)(t) = K \tag{19}$$

The controller $C$ that tracks the target value is designed as follows

$$\tau = -k_{\mathrm{p}}(q_2 - q_2^d) - k_{\mathrm{d}}\dot{q}_2 \tag{20}$$

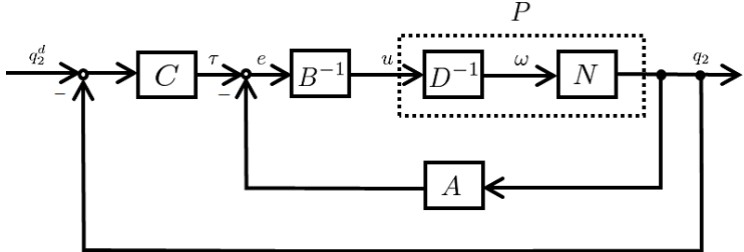

**Figure 5.** Nonlinear feedback system based on operator theory.

### 4.2.3. Robust Stability Condition

There are uncertainties in actual controlled objects that are difficult to express in mathematics. However, even if there are uncertainties, the robust stablity of the feedback system can be guaranteed if (21) is satisfied. We construct a nonlinear feedback system considering the uncertainties and robust stability. Figure 6 shows the nonlinear feedback system of the swing-up with uncertainties. In Figure 6, $P + \Delta P$ is the actual controlled object with uncertainties, where uncertainties are modelled for $D$ as $\Delta D$

$$||(B(D + \Delta D) - BD)M^{-1}||_{\mathrm{Lip}} < 1 \tag{21}$$

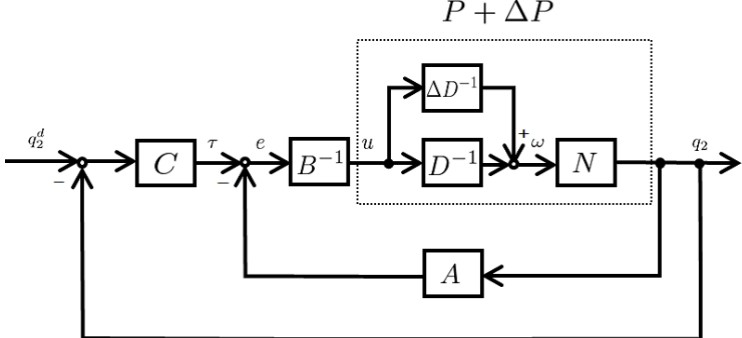

**Figure 6.** Nonlinear feedback system with uncertainties based on operator theory.

### 4.2.4. Control System Design Based on Operator Theory and Isomorphism Scheme

In this section, a control system design based on operator theory and an isomorphism scheme is employed for the swing-up. Suppose that the system $P + \Delta P$ is well-posed, using an appropriate isomorphism approach, the right coprime factorization of the unstable plant can be realized. The robust stability of the perturbed nonlinear feedback control systems can be guaranteed based on isomorphism. Besides, the plant output is able to track the reference input by the designed tracking controller [21]. The control system based on this isomorphism scheme can be described as a compensator $Q^{-1}$ and a feedback loop designed as shown in Figure 7. It then follows that the operator $\tilde{D}^{-1}$ is stable.

$$Q^{-1} = D(\tilde{D} - I) \tag{22}$$

$$N(\omega)(t) = K_1\omega(t) \tag{23}$$

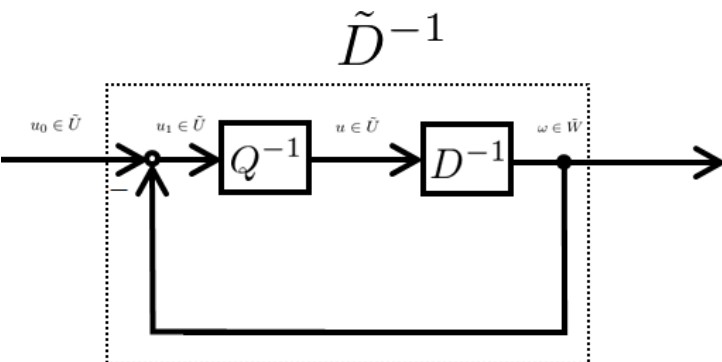

**Figure 7.** Nonlinear feedback loop based on an isomorphism scheme.

The compensator $Q^{-1}$ is designed in such a way that it stabilizes $D^{-1}$, the operators $N$ and $\tilde{D}^{-1}$ are equivalent operators, here, $K_1$ and $K_2$ are design parameters. The controllers $\tilde{A}$ and $\tilde{B}$ satisfying the Bezout equation $A\tilde{N} + \tilde{B}\tilde{D} = M$ are designed as follows. From (26), it can be confirmed that $\tilde{D}^{-1}$ is stabilized by the compensator $Q^{-1}$.

$$\tilde{A}(y)(t) = (1 - \frac{K_2}{K_1} y(t)) \tag{24}$$

$$\tilde{B}(u_0)(t) = K_2 u_0(t) \tag{25}$$

$$\tilde{D}^{-1}(u_0)(t) = K_1 u_0(t) \tag{26}$$

From the above analysis, we can see that there is a differential function in operator-based controller (18). However, by using our isomorphism scheme, it is avoided in (24). As a result, better control performance can be obtained. However, for the control system in Figure 8, when uncertainties exist, based on the argument in Section 4.2.3, the robust stability can be ensured.

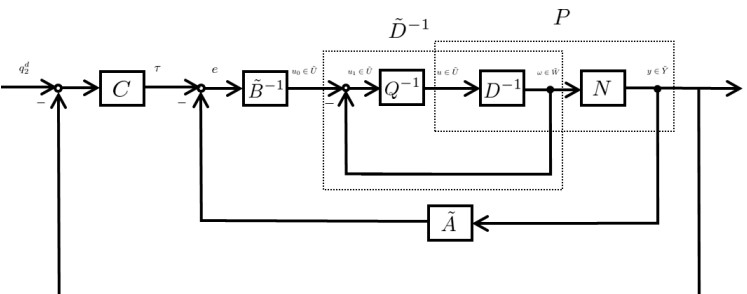

**Figure 8.** Nonlinear feedback system without uncertainties based on operator theory and an isomorphism scheme.

## 5. Simulation

In this section, we verify the effectiveness of the control system designed in Section 4 by simulation using MATLAB. Assuming that there are uncertainties in the angle and the magnitude is a constant with value 0.01 rad. Table 2 shows the mechanical parameters of simulation.

### 5.1. Control System Simulation Based on Operator Theory

In this section, we show some simulation results based on operator theory. In simulation, the control parameters are the following: $k_p = 200$, $k_d = 20$, $K = 350$, $K_1 = 1$, $K_2 = 1200$ and the initial states are $q_1 = -\frac{\pi}{2}$, $q_2 = 0$, $\dot{q}_1 = 0.3$ and $\dot{q}_2 = 0$. The simulation results of swing-up control and inverted stabilization control are shown in Figures 9–11. From Figure 12, the calculation result of the robust stability condition is always less than

1, and the robust stability of the control system is guaranteed. The simulation results of the conventional method and the proposed method are compared. The simulation results converge to $q_1 = \frac{\pi}{2}, q_2 = 0$, respectively. It can be seen from Figure 11 that the noise generated in the torque can be reduced when switching to inverted stability. It is clear that the swing-up control has been improved.

From the above, we are able to confirm the effectiveness of the proposed method.

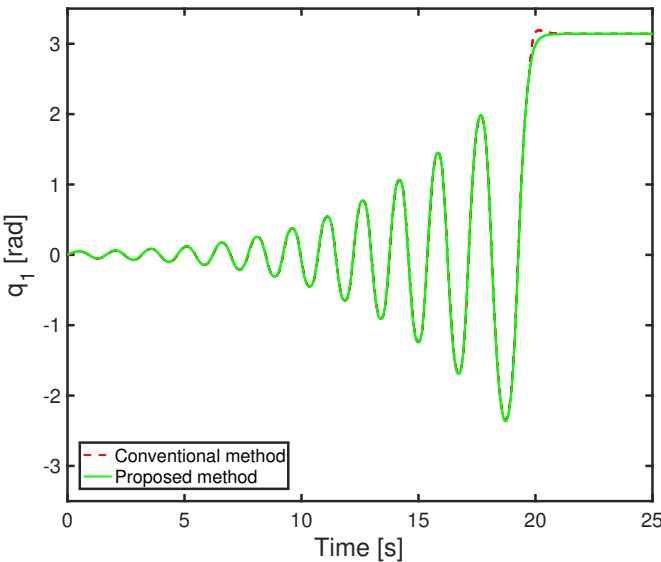

**Figure 9.** First link angle $q_1$ of nonlinear feedback system with uncertainties based on operator theory.

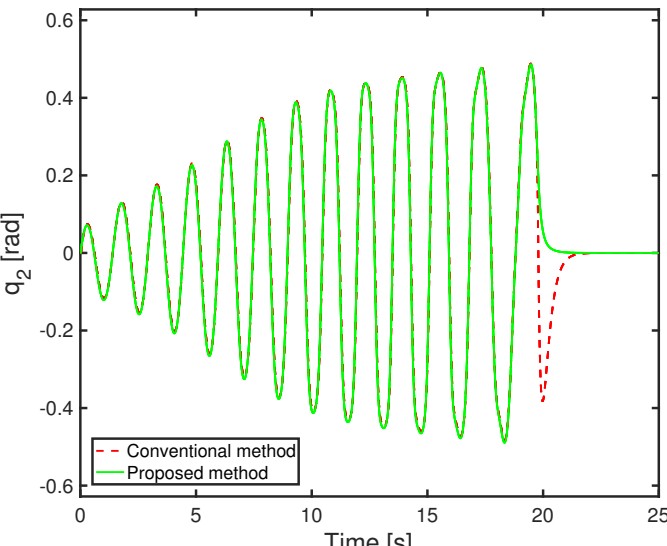

**Figure 10.** Second link angle $q_1$ of nonlinear feedback system with uncertainties based on operator theory.

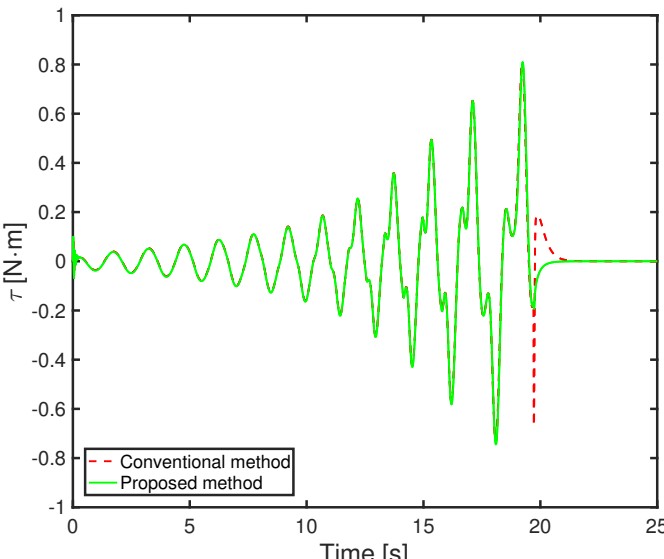

**Figure 11.** Input torque.

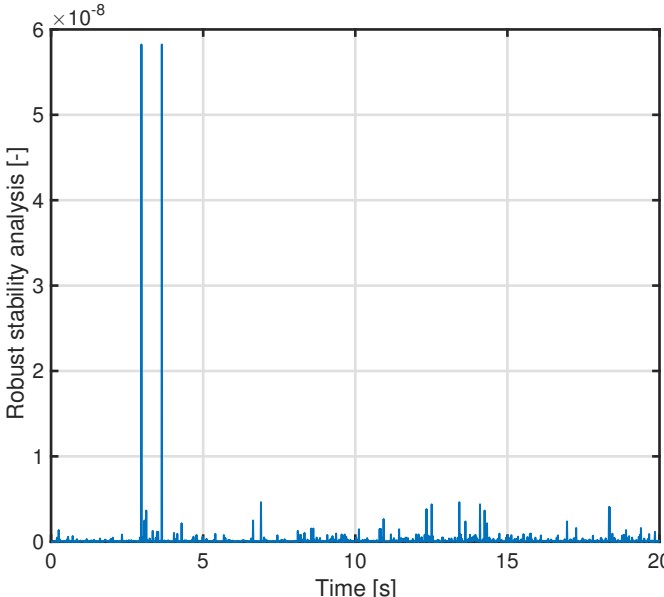

**Figure 12.** Robust stability assessment of nonlinear feedback system with uncertainties based on operator theory.

### 5.2. Control System Simulation Based on Operator Theory and Isomorphism Scheme

In this section, we show the simulation results based on operator theory and an isomorphism scheme (shown in Figures 12–16). In simulation, $K_1 = 1$, $K_2 = 1200$. Based on operator theory, in Figures 9 and 10, the response time is about 20 seconds. While based on operator theory and an isomorphism scheme, in Figures 13 and 14, the response time is about 8 s. The response time of the swing-up is reduced by 12 s after using our isomorphism scheme. If compared with [3], where the response time is around 18 s, the response time by using the proposed method is much shorter.

### 5.3. Robust Stability of a Control System

In this section, the robust stability of the control system is evaluated by calculating the Lipschitz norm (21). The simulation results are shown in Figures 12 and 15. In this simulation, it is assumed that there is uncertainty in the angle and its magnitude is a

constant with value 0.05 rad. From Figures 12 and 15, the robust stability conditions are always less than 1 and the robust stabilities of the control system are guaranteed.

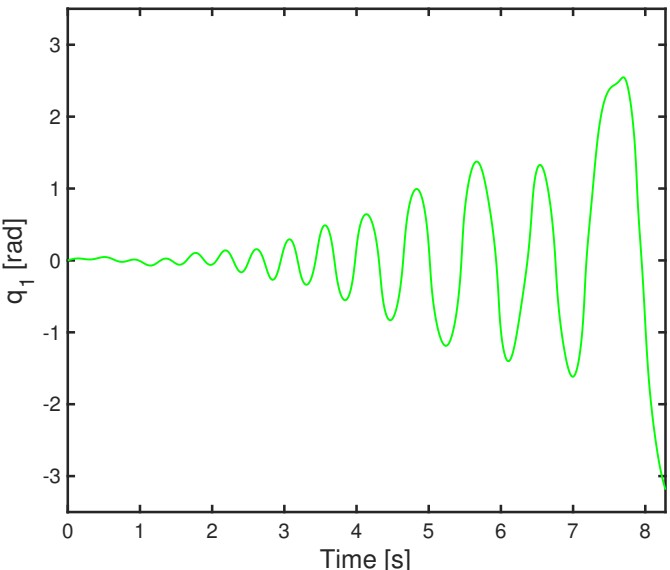

**Figure 13.** First link angle $q_1$ of nonlinear feedback system with uncertainties based on operator theory and isomorphism scheme.

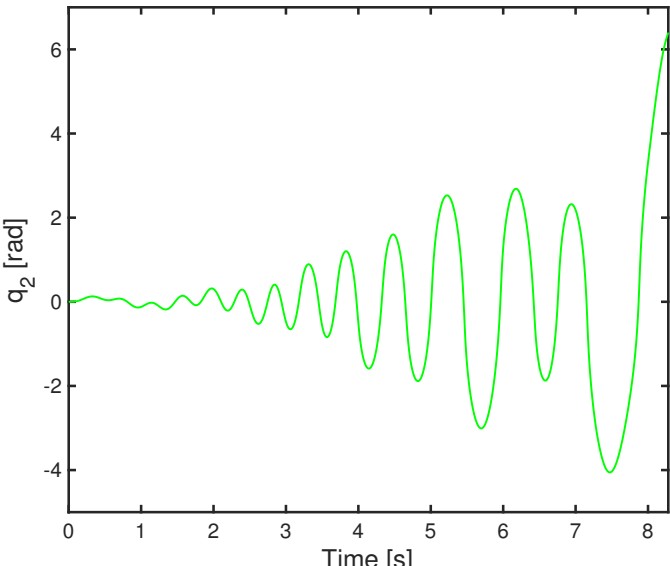

**Figure 14.** Second link angle $q_2$ of nonlinear control feedback system with uncertainties based on operator theory and an isomorphism scheme.

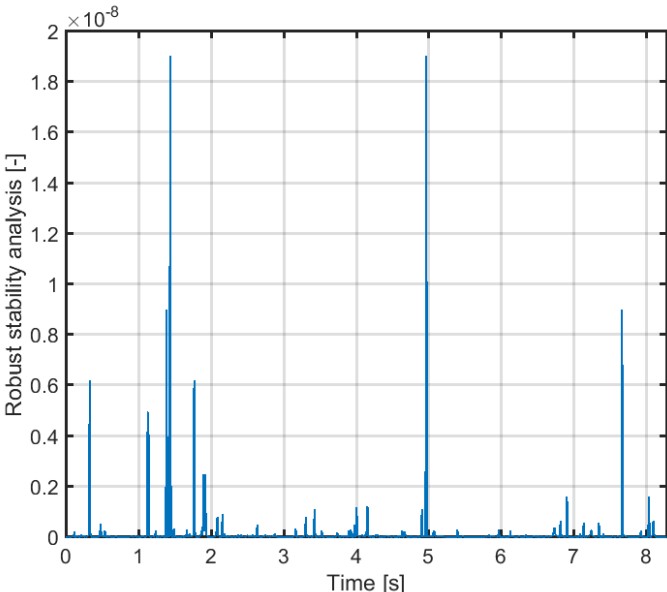

**Figure 15.** Robust stability assessment of nonlinear feedback system with uncertainties based on operator theory and isomorphism scheme.

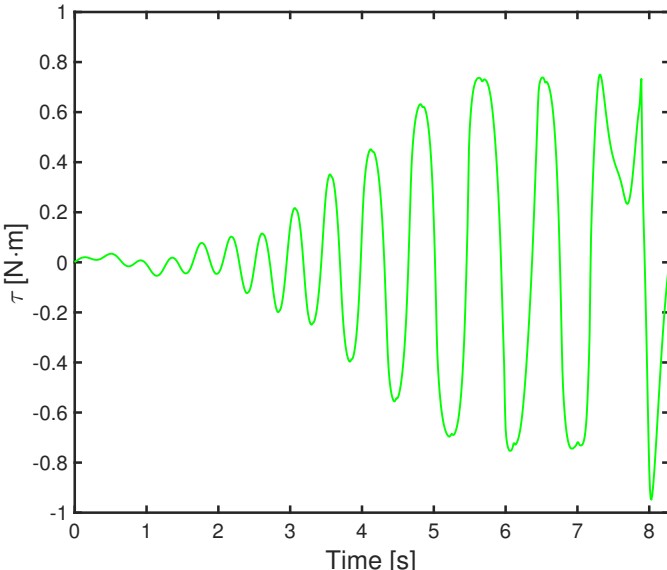

**Figure 16.** Input torque.

## 6. Conclusions

In this paper, a nonlinear control system of an underactuated robot based on operator theory and an isomorphism scheme is realized. Performances (robust stability, response time) of an underactuated robot nonlinear control system are also discussed. The effectiveness of the proposed method is verified by simulation. Optimizing the control parameters using some intelligent optimization methods [23,24] and adaptive learning methods [25–27]and further improving the tracking performance of an underactuated robot will be our future work. Besides, we will also perform experiments to verify the proposed method in the future.

**Author Contributions:** M.D. supervised the work and wrote the paper; S.K. finished simulation and the rest work. All authors have read and agreed to the published version of the manuscript.

**Funding:** This research received no external funding.

**Institutional Review Board Statement:** Not applicable.

**Informed Consent Statement:** Not applicable.

**Data Availability Statement:** Not applicable.

**Acknowledgments:** The authors would like to thank PhD students Guanqiang Dong and Yuanhong Xu at Tokyo University of Agriculture and Technology, Japan, for their suggestions and comments.

**Conflicts of Interest:** The authors declare no conflict of interest.

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
