# Peer review of "Nonlinear Control System Design of an Underactuated Robot Based on Operator Theory and Isomorphism Scheme"

_axioms, doi:10.3390/axioms10020062_

Round 1

Reviewer 1 Report

There are several minor issues to be taken care of. The details can be found in the referee report.

Author Response

Thanks.

Reviewer 2 Report

Please see the attached file. I hope that my comments and suggestions help you to improve the paper.

Author Response

Thanks.

Reviewer 3 Report

Dear authors, the subject of research in the submitted article is acrobot. This is a well-known example of an underactuated system. The proposition of a method of design a control system based on operator theory and an isomorphism scheme is the main aim of the submitted paper. The presented method is interesting, but the article itself requires significant corrections .

First of all, the introduction should be more elaborate. There is no objective literature review. The bibliography contains only 15 items, the vast majority of which are the authors' own works. There is no way to show how other researchers approach this issue. General information about the control of the class of underactuated mechanical systems may be also provided. 

The conclusions are also not convincing. The results obtained are not better than the acrobot control systems developed by other research teams. The stabilization time is quite long. Justify why the presented method should be worth implementing. 

Also, please verify the notations in equations 15 and 20. There, the parameter Kd is multiplied by the generalized variable q2. The diagram in Figure 5 shows that this is the standard concept of a feedback loop, so the parameter Kd should be multiplied by the error derivative (q-qd). If not, please explain it in detail.

In my opinion, the article cannot be published in this version. 

Author Response

Thanks.

Round 2

Reviewer 2 Report

Thank You to consider all of my comments and suggestions. At this time, I think that the paper is ready to be accepted in present form.
Best regards

Reviewer 3 Report

Thank you for the answers. I think the article can be published in this form.